# The Preclinical Validation of 405 nm Light Parasiticidal Efficacy on *Leishmania donovani* in Ex Vivo Platelets in a Rag2^−/−^ Mouse Model

**DOI:** 10.3390/microorganisms12020280

**Published:** 2024-01-29

**Authors:** Pravin R. Kaldhone, Nazli Azodi, Hannah L. Markle, Neetu Dahiya, Caitlin Stewart, John Anderson, Scott MacGregor, Michelle Maclean, Hira L. Nakhasi, Sreenivas Gannavaram, Chintamani Atreya

**Affiliations:** 1Division of Blood Components and Devices, Center for Biologics Evaluation and Research, Food and Drug Administration, Silver Spring, MD 20993, USA; pravin.kaldhone@fda.hhs.gov (P.R.K.); neetu.dahiya@fda.hhs.gov (N.D.); 2Division of Emerging and Transfusion Transmitted Diseases, Center for Biologics Evaluation and Research, Food and Drug Administration, Silver Spring, MD 20993, USA; nazli.azodi@fda.hhs.gov (N.A.); hannah.markle@fda.hhs.gov (H.L.M.); hira.nakhasi@fda.hhs.gov (H.L.N.); 3Department of Electronic and Electrical Engineering, University of Strathclyde, Glasgow G1 1XW, UK; caitlin.stewart@strath.ac.uk (C.S.); j.g.anderson@strath.ac.uk (J.A.); scott.macgregor@strath.ac.uk (S.M.); michelle.maclean@strath.ac.uk (M.M.); 4The Robertson Trust Laboratory for Electronic Sterilization Technologies, Department of Electronic and Electrical Engineering, University of Strathclyde, Glasgow G1 1XW, UK

**Keywords:** 405 nm light, platelets, *Leishmania*, pathogen reduction, transfusion transmissible parasites, transfusion safety

## Abstract

Violet–blue light of 405 nm in the visible spectrum at a dose of 270 J/cm^2^ alone has been shown to be an effective microbicidal tool for inactivating several bacteria, HIV-1, and *Trypanosoma cruzi* in ex vivo plasma and platelets. Unlike chemical- and ultraviolet (UV)-based pathogen inactivation methods for plasma and platelet safety, 405 nm light is shown to be less toxic to host cells at light doses that are microbicidal. In this report, we evaluated the parasiticidal activity of a 405 nm light treatment on platelets spiked with the *Leishmania donovani* parasite. Following the light treatment, parasite viability was observed to be near zero in both low- and high-titer-spiked platelets relative to controls. Furthermore, to test the residual infectivity after inactivation in vivo, the light-treated low-titer *L. donovani*-spiked platelets were evaluated in an immunodeficient Rag2^−/−^ mouse model and monitored for 9 weeks. The parasiticidal efficacy of 405 nm light was evident from the lack of a presence of parasites in the mice spleens. Parasiticidal activity was confirmed to be mediated through 405 nm light-induced reactive oxygen species (ROS), as quantitatively measured by a 2′,7′-Dichlorodihydrofluorescein diacetate (H_2_DCFDA)-based assay. Overall, these results confirm the complete inactivation of *L. donovani* spiked in ex vivo platelets by 405 nm light treatment and exemplify the utility of the Rag2^−/−^ mouse infection model for the preclinical validation of the parasiticidal efficacy of 405 nm light and this light-based technology as a potential PRT for ex vivo platelets.

## 1. Introduction

Donor screening prior to collection and post-collection testing of ex vivo blood and blood components stored for transfusion dramatically reduces the risk of contamination of blood-borne pathogens in the recipients. Nonetheless, this risk can never be completely eliminated due to the limitations of the testing technologies employed. With the introduction of chemical and UV light-based pathogen reduction technologies (PRTs) for individual blood components (plasma, platelets, and red blood cells), the risk has been further mitigated [1]. While PRTs currently in use for blood safety are all effective against blood-borne pathogens, they do impede the full functional spectrum of platelets and plasma-coagulation factors, resulting in the lower end of the acceptable transfusion efficacy of the treated products. Specific protein kinase signaling pathways have been reported to be affected or induced in platelets upon treatment with current UV-based PRTs [2]. Towards circumventing the harmful effects of UV light, we evaluated a less harmful visible spectrum violet–blue light of a 405 nm wavelength as an alternative to UV light for the microbicidal treatment of ex vivo platelets and plasma in order to ensure safety from transfusion-transmitted infections (TTIs). Especially since platelets are stored at room temperature, they are vulnerable to microbial contamination and subsequent growth; it is important to evaluate the microbicidal effect of 405 nm light in platelets stored in plasma. We have shown that 405 nm light is a potent pathogen inactivator for a number of bacteria, HIV-1, and a parasite *Trypanosoma cruzi*, while gentle on the treated human plasma and platelets stored for transfusion [3,4,5,6,7,8,9].

Currently there are no active surveillance measures nor an FDA-approved donor screening assay to detect *Leishmania* in ex vivo blood components stored for transfusion, including platelet concentrates (PCs). Autochthonous cases of leishmaniasis were reported in Southern US [10], and leishmaniasis is notifiable only in the state of Texas. Therefore, *Leishmania* has been considered endemic to the United States since 2015. Studies of a *Leishmania* infection epidemiology in the U.S. are limited to the surveillance of armed forces members, case reports, and multicenter observational studies [11]. Since the majority of *Leishmania* infections result in asymptomatic cases [10], inactivating this parasite in platelets through PRTs is a more practical way to mitigate the TTI risk. Moreover, especially in military settings, PCs are life savers for severe trauma and bleeding injuries. Hence, we assessed the parasiticidal activity of 405 nm light on *L. donovani* spiked in PCs.

To test the in vitro efficacy of 405 nm light against *L. donovani* parasites, platelets spiked with parasites were either treated with violet–blue light or left untreated and counted under a microscope to determine parasite viability. To confirm that the 405 nm light parasiticidal activity was imparted through ROS induction, after light treatment, parasite viability was measured via flow cytometry and a fluorogenic 2′-7′-dichlorofluoprescin diacetate (H_2_DCFDA) assay to quantitate ROS via release of H_2_O_2_ in live samples. Since parasite scores through microscopic examination was not a reliable measure for parasite viability, in vivo analyses of the parasiticidal efficacy of the 405 nm treatment were conducted by transferring treated and untreated parasite-spiked platelets into the B- and T-cell deficient Rag2^−/−^ mice, mimicking the human population of immunocompromised transfusion recipients. The susceptibility of Rag2^−/−^ mice to an *L. donovani* infection has been shown previously [12,13]. Our results revealed that 405 nm exposure to *L. donovani*-spiked platelets effectively and completely inactivates parasites. These preclinical studies validate the efficacy of 405 nm violet–blue light and support its utilization as a PRT for *Leishmania* parasites and the Rag2^−/−^ mouse model for the confirmation of parasite inactivation.

## 2. Materials and Methods

### 2.1. Human Platelets

Human platelets suspended in plasma were obtained from the National Institute of Health Clinical Center Blood Bank (Bethesda, MD, USA). Platelets were stored at 22 ± 2 °C under gentle agitation for up to one day. The FDA Research Involving Human Subjects Committee approved this protocol (RIHSC, Exemption Approval #11-036B).

### 2.2. Parasites

Cryopreserved *L. donovani* parasites were cultured at 27 °C in an M199 medium supplemented with 10% heat-inactivated fetal bovine serum, 0.4% adenosine (250×), 0.1% folic acid, 1% penicillin/streptomycin, 1% glutamine, 1% BME vitamins, Hepes powder, and sodium bicarbonate and then routinely passaged. Although amastigote parasites are more likely to be found in a potentially infected donor platelet preparation, axenic amastigotes are only possible to be cultured from *Leishmania donovani* parasites collected from infected mice. While it is possible to purify amastigotes from infected mice, mice-derived amastigotes may not be pure enough and may contain material of mouse origin that could affect the efficacy of 405 nm exposure. Similarly, spiking in vitro infected human monocyte-derived macrophages to the platelets prior to 405 light exposures may not be suitable since controlling for parasite titers in the infected macrophages is a challenge. Considering these variables, we decided to use axenic promastigotes in our experiments, and we recognize that this is a limitation of this study.

### 2.3. The Delivery of 405 nm Violet–Blue Light

The 405 nm light treatment of platelets was performed in a closed system (US Patent Application no. 62/236, 706, 2015) as previously reported (7,3,8). Briefly, the system contained a light source consisting of narrowband 405 nm LED arrays (FWHM ~20 nm; LED Engin, San Jose, CA, USA), with appropriate thermal management, and powered by LED drivers (Mean Well, Xinbei, Taiwan) and a shaker platform to have 60 rpm and 22 °C in order to provide the continuous agitation of the sample while maintaining consistent treatment conditions. The distance of 14 cm was held between the light source and the samples. For the 405 nm light treatment of platelets spiked with *Leishmania*, samples were placed in T-150 Terumo Transfer bags (Terumo BCT Inc., Lakewood, CA, USA) with a volume of 40 mL (n = 6). The irradiance of 405 nm light across the area of the samples was anticipated to be approximately 15 mW/cm^2^ [3]. The samples were treated with 405 nm light for 5 h. The light treatment dose (Jcm^−2^) was calculated by multiplying the irradiance (W/cm^2^) and duration of the treatment. Hence, the dose level after 5 h of 405 nm light treatment at an irradiance of ~15 mW/cm^2^ was equivalent to 270 J/cm^2^.

### 2.4. Animals Used in the Study

The animal protocol for this study has been approved by the Institutional Animal Care and Use Committee at the Center for Biologics Evaluation and Research, US Food and Drug Administration (FDA) (ASP 1995#26). In addition, the animal protocol is in complete accordance with “The guide for the care and use of animals as described in the US Public Health Service policy on Humane Care and Use of Laboratory Animals 2015”. Female B6.Cg-Rag2tm1.1Cgn/J (Rag2^−/−^) mice were purchased from Jackson Laboratories (Bar Harbor, ME, USA) and housed in the FDA animal facility at one mouse per cage and provided with autoclaved water. Mice aged 4–6 weeks were randomly assigned to one of four experimental groups: Group 1: 405 nm-treated *L. donovani*-spiked platelets, Group 2: untreated *L. donovani*-spiked platelets, Group 3: *L. donovani* control, and Group 4: naïve control.

### 2.5. The Detection of In Vitro Parasite Viability Following 405 nm Treatment

Human platelets suspended in plasma from six donors were spiked with low-titer (10^4^ parasites/mL) *L. donovani* parasites counted with an automated cell counter in either glass flasks or platelet donation bags and randomly assigned as treated or untreated. Platelets marked as treated were exposed to 405 nm violet–blue light for 5 h to achieve the light dose of 270 J/cm^2^, while untreated platelets were light-protected by covering with aluminum foil and incubated along with the light-treated platelets. At 0 and 5 h post-treatment times, both the treated and untreated platelets were diluted 10-fold in PBS, and a final volume of 10 µL was used to manually count with a hemocytometer for a total of 10 repeat counts. The same method was also repeated with high-titer (10^7^ parasites/mL)-spiked platelets.

### 2.6. H_2_DCFDA Fluorogenic Assay to Detect ROS

Cryopreserved *L. donovani* parasites were cultured as described above and subsequently passaged as needed. The platelets were spiked with *L. donovani* (2 × 10^7^ parasites/mL) and divided into two T25 Nunc flasks of 5 mL each, one treated with 405 nm violet–blue light for 5 h and the other wrapped in tin foil to prevent light exposure, being left untreated. In addition, the *L. donovani* parasite culture was split into another two flasks, 5 mL each, and one flask was used for 405 nm light treatment while the other was used as an untreated control. After the treatment, the samples were dispensed into a 96-well plate. The H_2_DCFDA reaction was set up by adding a 20 µM H_2_DCFDA compound (Thermo Fisher Scientific Inc., Waltham, MA, USA) and incubating at 37 °C for 30 min. Ten replicates were set up for each group. Hydrogen peroxide at 0.5 µM was used as a positive control, and fresh culture of *L. donovani* in media was used as a negative control. After the incubation, fluorescence was measured with spectrophotometer (Spectramax^®^ iD5, Molecular Devices LLC., San Jose, CA, USA) at an excitation/emission wavelength of 492/532. The readings were acquired and analyzed through SoftMax Pro (version 7.1) software.

### 2.7. Analysis of Live/Dead Cells by Flow Cytometry Following Treatment with 405 nm Light

*L. donovani* promastigotes were acquired on a Cytek Aurora analyzer equipped with UV, Violet, Blue, Yellow–Green, and Red lasers using SpectroFlo software v3.0.3. Data were analyzed with the FlowJo software v10.7 (Becton Dickinson Inc., San Jose, CA, USA). To detect parasites on the flow cytometer, voltages were optimized using 4 μm size calibration Aldehyde/Sulfate Latex Beads, 4% *w*/*v* (Invitrogen Corp., Waltham, MA, USA). Doublets were removed using width parameters. Dead cells were identified based on staining with LIVE/DEAD™ Fixable Aqua Stain (Life Technologies, Frederick, MD, USA).

### 2.8. In Vivo Analysis of the 405 nm Light Inactivation Efficacy of Parasites in Rag2^−/−^ Mice

Human platelet products were spiked with *L. donovani* promastigotes (10^4^ parasites/mL), counted with an automated cell counter, and treated with 405 nm violet–blue light for 5 h or covered with aluminum foil and left untreated. Rag2^−/−^ mice were inoculated intravenously with 100 µL of the treated (N = 5) or untreated (N = 5) parasite-spiked platelets at a concentration of 10^3^ parasites/mL and sacrificed at 9 weeks post-inoculation. Rag2^−/−^ mice (N = 4), inoculated intravenously with 10^3^ parasites, were used as a positive control and sacrificed at 7 weeks post-inoculation. One mouse was used as a naïve control. The spleen and bilateral cervical lymph nodes were collected from each mouse, macerated with a syringe plunger and a 70 µm cell strainer, and cultured in a 96-well plate in 1:10 serial dilutions in an M199 medium supplemented with 0.1% gentamicin and 0.2% 6-biopterin. After two weeks, parasite burden by the limiting dilution method was reported as a log titer of the maximum dilution where parasites were observed.

### 2.9. Statistical Analysis

A paired *t*-test with a significance threshold of 0.05 was used to analyze pooled data of in vitro manual parasite viability counts in low- and high-titer parasite-spiked donor platelets. Individual counts compromising the pooled data above conducted on separate experimental dates were analyzed for statistical significance via one-way ANOVA tests with a significance threshold of 0.05. The fluorescence readings were compared through a two-tailed paired *t*-test with a significance threshold of 0.05 (GraphPad Prism, version 9.5.0). For parasite burden, statistical significance was calculated using a paired *t*-test (*p* < 0.0001).

## 3. Results

### 3.1. Parasite Viability Decreased In Vitro Following 405 nm Light Exposure

Parasite viability in both 405 nm treated and untreated groups spiked with low- and high-titer parasite concentrations via microscopic examination and manual counting was utilized as the initial measure of 405 nm efficacy against *L. donovani* parasites. A notable loss in parasite mobility (=viability) was observed in both low- and high-titer parasite-spiked samples following 5 h exposure to 405 nm light. As seen in Figure 1, parasite viability counts decreased significantly after 5 h of light treatment (=270 J/cm^2^ light dose) in both the low- (Figure 1A) and high- (Figure 1B) titer parasite groups. We observed that the light-untreated group that was kept in dark at 22 °C on a 60 rpm platform for 5 h (same conditions as for the light-treated samples) also showed a slightly lower viability compared to the parasite-alone controls, suggesting that these conditions are not conducive for parasites to thrive. At the 0 h timepoint, for each count in both the treated and untreated, low-, and high-titer groups, the parasites were motile, elongated, and clearly viable. At the 5 h timepoint, following the blue light treatment, parasite morphology remained largely unchanged, with very few parasites displaying as more rounded and less elongated, and, notably, parasite motility drastically decreased or ceased altogether, which is indicative of the loss of viability. Given the different levels of opacity and thickness between glass flasks and platelet donation bags, as well as the potential of variable light penetration through PC containers, the counts were repeated multiple times in both glass flasks and donation bags and arrived at similar parasite kill numbers in both scenarios.

### 3.2. 405 nm Light Treatment Was Associated with Reactive Oxygen Species Altered Parasite Viability

Previous studies have shown that exposure to 405 nm light induces ROS from the bacterial cell substrates, such as porphyrins, that lead to cell death [9]. In order to correlate 405 nm light-associated parasite inactivation with the light-induced ROS, following in vitro parasite viability counts, another assay to detect the potential mechanism of action of parasite injury was conducted. The H_2_DCFDA assay, using the fluorogenic, cell-permeable H_2_DCFDA reagent, allows for a quantitative output of ROS in live cells. As seen in Figure 2A, ROS was produced as a byproduct of 405 nm light-mediated parasite destruction. Fluorescence, represented as a relative fluorescence unit (RFU), from the 405 nm light-treated groups was higher than untreated group. This difference is statistically significant. Both parasites and platelets spiked with parasites in the treated group had higher RFU counts than the respective untreated groups. Flow cytometry results depicted clear differences in the percentage of dead cells per sample population, as shown in Figure 2B. The percentages of dead parasites were noted to be significantly higher in the treated samples as compared to the untreated ones, as shown in Figure 2C.

### 3.3. 405 nm Light-Treated Parasite Infectivity Was Diminished In Vivo in Rag2^−/−^ Mice

To test the in vivo infectivity of the parasites following 405 nm light exposure, 100 uL of the human plasma containing 10^4^/mL parasites (low titer) was intravenously injected into Rag2^−/−^ mice. Microscopic counts of the parasites prior to inoculation into Rag2^−/−^ mice showed that a significant reduction in viable parasite counts was achieved following 405 nm light exposure (Figure 3A). Stationary phase promastigotes of virulent *L. donovani* parasites [14] were used as a positive control (Figure 3B). Parasite burdens in spleens assessed 7–9 weeks following inoculation showed that no viable parasites were recoverable in the 405 nm light-treated group, whereas in both untreated and parasite-alone control groups, viable parasites were present at significantly higher levels (Figure 3C). Consistent with our in vitro data, untreated groups did show a slight reduction in splenic parasite burdens compared to the parasite-alone group (Figure 3C). This was expected since the untreated group, i.e., the group shielded from 405 nm exposure, demonstrated slightly lower viability due to the intrinsic microbicidal activity of platelets compared to parasite-alone controls cultured in ideal conditions for growth in incubators.

## 4. Discussion

Toward the goal of decreasing transfusion-transmitted diseases, several pathogen reduction technologies have been developed and tested. In the past, solvent/detergent and dye-based systems have been used for pathogen inactivation purposes but were withdrawn due to significant damage to platelets [15]. Among the recent PRTs, INTERCEPT^TM^ (Cerus Corp., Concord, CA, USA) technology uses synthetic psoralen S-59, DNA intercalator [16], while Mirasol^®^ (Terumo Inc., Lakewood, CA, USA) technology involves the addition of riboflavin and ultraviolet (UV) light [15,17,18]. These technologies require the addition of chelators, and sample exposure to UV light has been reported to reduce the in vitro quality of platelets [15], which calls for alternative methods that are gentler to the platelets. We have previously shown that 405 nm light is a potent pathogen inactivator for a number of bacteria, HIV-1, and the *Trypanosoma cruzi* parasite, while it is gentle on treated plasma and platelets in human plasma and on platelets stored for transfusion [3,4,5,6,7,8]. Platelets treated with 405 nm light behaved similar to the light-untreated platelets in terms of in vitro parameters such as pH, lactate release, activation/aggregation responses to collagen, and the in vivo response of survival and recovery in an immunodeficient SCID mouse model [3,7]. Furthermore, we also demonstrated that protein integrity was not compromised in 405 nm light-treated plasma [9]. Hence, we believed 405 nm light was compatible with platelets and plasma and continued our studies. Here, in this report, 405 nm light in the visible spectrum was evaluated for Leishmania parasite inactivation in stored platelets.

The safety of 405 nm light and its microbicidal properties has led to its utility in a range of settings, including for air purification, hospital decontamination, and even direct application to skin and wounds as antimicrobial treatments [19]. For example, in vivo studies of methicillin-resistant *Staphylococcus aureus* (MRSA) and *Pseudomonas aeruginosa* have shown dramatic reductions in bacterial load following the direct application of blue light to wounds and burns. In these studies, the host tissues were exposed to blue light to directly inactivate pathogens within open wounds and burns [20,21]. While the wavelengths of blue light slightly differ from what was used in our studies (415 nm versus 405 nm), the concept of direct blue light application leading to pathogen destruction can be translated to our studies. Furthermore, these studies implicate the safety of blue light treatment, as direct energy was applied to living host tissue with no harmful side effects noted [20,21]. The microbicidal efficacy of 405 nm light has been shown for various pathogens so far. Ragupathy et al. have shown that 405 nm light can inactivate HIV-1 in human plasma [8]. Similarly, Jankowska et al. noted the ability of 405 nm light to inactivate the *Trypanosoma cruzi* parasite present in human platelets [3].

Safety characteristics of various PRT technologies are typically assessed in in vitro studies using platelets spiked with various microbial agents [22,23]. In these testing modalities, very high titers of microbes are spiked into human blood products, and several log-reduction effects are typically reported following the treatment as a marker of the efficacy of the PRT employed. The testing of blood products spiked with a low titer of microbes is more practical since it reflects latent infections more accurately. Latent infections of *Leishmania*, for instance, are often PCR- negative and require either a serology or the intradermal inoculation of leishmanin antigens, which induces a delayed type-hypersensitivity reaction, suggesting a previous exposure akin to a Tuberculin test [9,24]. Developing animal models that enable the rigorous evaluation of the efficacy of various PRTs under physiologically relevant microbe titers is therefore necessary. Previous studies employed immunodeficient mouse models to assess transfusion-associated GVHD [14] but not the microbicidal efficacy of PRTs. The persistence of latent infections of *Leishmania* is often assessed in various immunodeficient mouse models and under dexamethasone-induced deficiency [25,26]. On the other hand, the Rag2^−/−^ mouse model, lacking both mature T and B cells, represents the best method to assess microbicidal activity under a low-titer spiking regimen to test the efficacy of 405 nm treatment against not only *Leishmania* parasites as shown in this study but also a broad range of transfusion-transmissible microbes.

Given the safety benefits and decreased toxicity compared to other methods of light exposure, 405 nm violet–blue light is a key tool in the reduction of pathogens within PCs. The notable decrease in *Leishmania* parasites showcased here in both in vitro and in vivo scenarios and previous reports from this group on the inactivation of other microbes support the potential application of 405 nm violet–blue light treatment as concerns the safety of PCs. Specifically, the Rag2^−/−^ immunocompromised mouse model mimics the immunocompromised human patient population receiving PCs, wherein there is an ongoing concern of transfusion-transmitted infections due to a lack of a defense against transfusion-transmitted pathogens. However, more work and rigorous testing of 405 nm light to optimize this methodology is highly warranted. This may include 405 nm treatment on pathogens that are currently understudied, refining the time of exposure or the amount of energy delivered, etc., as a future endeavor. A final note to our observations is that the parasites, which were kept in dark at 22 °C on a 60 rpm platform to serve as a control group, also demonstrated a slight loss of viability even without light treatment, suggesting that certain in vitro environmental conditions can play an adverse role to parasite viability.

## 5. Patents

A patent is pending on the blue light apparatus used in this study (U.S. Patent Application number 62/326, 706, 2015).

## Figures and Tables

**Figure 1 microorganisms-12-00280-f001:**
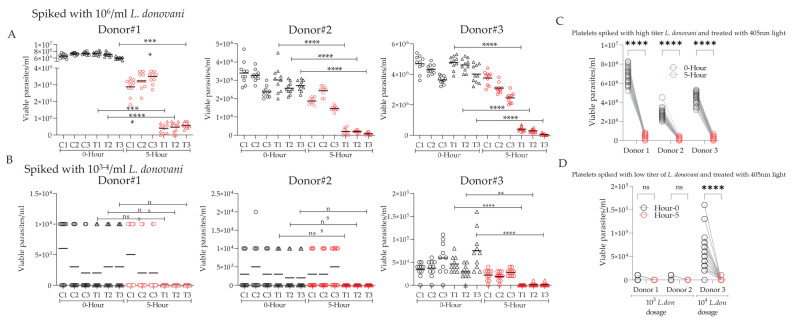
In vitro parasite viability counts pre- and post-405 nm light exposure. Six individual donor platelets suspended in plasma were spiked with high- (**A**) and low- (**B**) titer *L. donovani* promastigotes. For each donor, three spiked plasma bags were treated for five hours with 405 nm light, while three remained untreated. The parasitic viability of the treated and untreated samples was measured via microscopic examination and manual counting via a hemocytometer at 0 and 5 h post-405 nm treatment. These counts conducted on separate experimental dates were analyzed for statistical significance via one-way ANOVA tests with a significance threshold of 0.05. The parasitic viability counts for the high- and low-titer *L. donovani*-spiked donor platelets are pooled and depicted in (**C**,**D**), respectively. A paired *t*-test with a significance threshold of 0.05 was used to analyze the pooled data (ns, non-significant, * *p* < 0.05, ** *p* < 0.01, *** *p* < 0.001, **** *p* < 0.0001).

**Figure 2 microorganisms-12-00280-f002:**
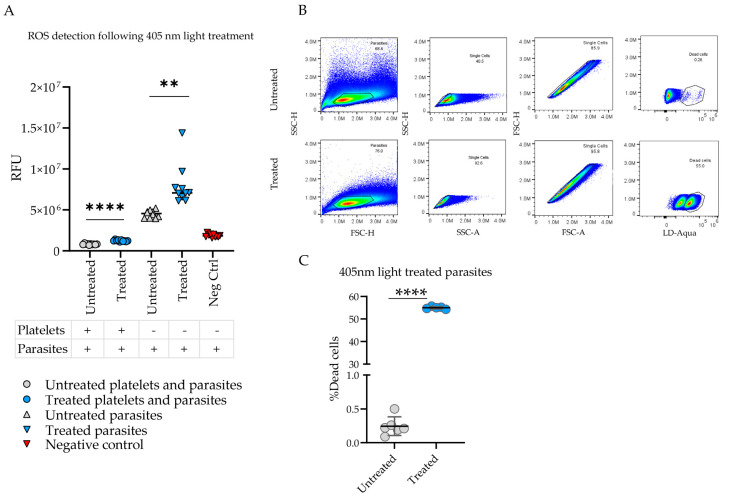
405 nm light treatment generates reactive oxygen species and is responsible for parasite inactivation. (**A**) Reactive oxygen species (ROS) detection via a H2DCFDA fluorometric assay in platelet and/or *L. donovani* parasite samples in 405 nm light-treated groups (blue) versus untreated groups (red), as measured by relative fluorescence units (RFU). Statistical significance between the observed values was calculated by a two-tailed paired *t*-test with a significance threshold of 0.05. (**B**) Flow cytometry pseudocolor plots representative of the gating strategy to detect parasite viability in untreated versus 405 nm light-treated samples. Viability was determined by staining cells with Life Technologies LIVE/DEAD Fixable Aqua Stain, removing doublets, and gating the dead cell population. (**C**) Parasite viability outcomes via flow cytometric analysis of untreated (gray) versus 405 nm light-treated (blue) samples, measured by the percentage of dead cells. Statistical significance observed between samples was calculated by a two-tailed paired *t*-test with a significance threshold of 0.05 (** *p* < 0.01, **** *p* < 0.0001).

**Figure 3 microorganisms-12-00280-f003:**
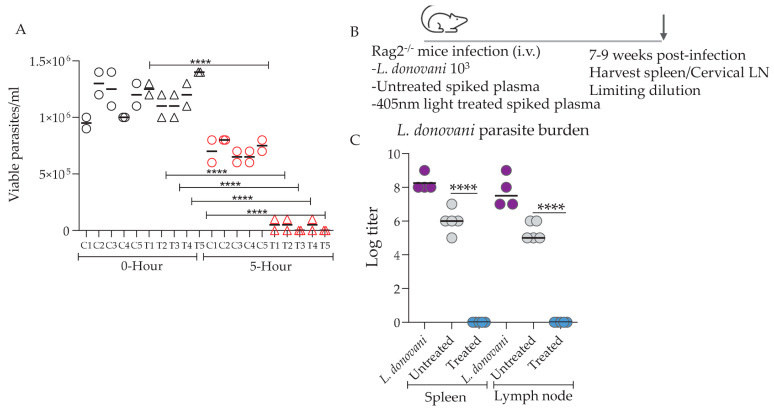
Parasite burden following the in vivo transfer of 405 nm treated and untreated *L. donovani*-spiked platelets. (**A**) Viable parasite counts by microscopy following 405 nm light treatment are shown. (**B**) Rag2^−/−^ mice were inoculated with 100 µL of human plasma spiked with *L. donovani* promastigote at a concentration of 104/mL. *L. donovani* promastigotes (103, i.v.) in 100 µL PBS were used as a positive control. A total of 100 µL of treated or untreated spiked plasma was inoculated into the mice. Spleens and superficial cervical lymph nodes were isolated from the positive controls (7 weeks p.i., indicated by the arrow) and the spiked human plasma- inoculated mice (9 weeks p.i.). (**C**) Parasite burden was estimated by using a limiting dilution assay. The parasite numbers are expressed as log2 titers observed after 3 weeks of culture. Statistical significance between the observed values was calculated by a paired *t*-test (**** *p* < 0.0001).

## Data Availability

The original contributions presented in this study are included in the article; further inquiries can be directed to the corresponding author.

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
