# Peer review of "The Preclinical Validation of 405 nm Light Parasiticidal Efficacy on Leishmania donovani in Ex Vivo Platelets in a Rag2−/− Mouse Model"

_microorganisms, 2024, doi:10.3390/microorganisms12020280_

Round 1

Reviewer 1 Report

Comments and Suggestions for Authors

In the manuscript: "Preclinical Validation of the Parasiticidal Effectiveness of 405 nm Light on Leishmania donovani in Ex Vivo Platelets in a Rag2-/- Mouse Model," the authors assess the parasiticidal impact of treating platelets spiked with Leishmania donovani parasite with 405 nm light. They examine changes in parasite viability, the production of reactive oxygen species, and the infectivity of the parasites after exposure to 405 nm light.

After conducting a thorough examination of the research, it is suggested that the authors make modifications to the following aspects:

1.       In the article's introduction, the provided information emphasizes the need to find effective methods against pathogens in transfused blood. It lists the drawbacks of existing methods and refers to the authors' previous research with 405 nm light on bacteria, HIV-1, and Trypanosoma cruzi. However, the authors do not explicitly clarify the importance of assessing the parasiticidal effect of 405 nm light specifically in platelets, rather than evaluating it in the complete blood from donors.

2.       In relation to point 1, under normal physiological conditions, it is anticipated that the blood of patients will only contain Leishmania promastigote forms during the initial hours post-infection. Beyond this timeframe, the primary form of the parasite in the bloodstream is expected to be the amastigote form, present both within phagocytic cells and released into the blood upon the rupture of the host cell. Additionally, various forms of the parasite may demonstrate varying susceptibility to chemical and/or physical treatments. For example, it is expected that the amastigote form within phagocytic cells may be less susceptible to environmental damage.

he authors are recommended to provide a rationale for their choice of the promastigote form for inoculation in platelet samples.

Additionally, it is suggested that, in the "Materials and Methods" section, the authors explicitly state that the Leishmania donovani stage utilized in the experiments was the promastigote form.

3.       The authors are encouraged to incorporate citations from the literature that support the specified samples treatment conditions outlined in section 2.3.

4.       Throughout the study, the authors specify the quantity of parasites per milliliter introduced into each platelet sample. However, it is not clarified whether the platelet concentration and/or the platelet-parasite ratio remained constant across all samples. This is critical for two reasons: a) a higher concentration of particles/cells in the sample may influence the penetrability and, consequently, the effectiveness of 405 nm light on the parasite; b) variations in the platelet-parasite ratio among samples could, to some extent, account for the observed differences in parasite cell viability in Figure 1C, as the researchers report consistent results indicating the microbicidal activity of platelets against L. donovani.

5.       In section "2.5. Detection of in vitro parasite viability following 405 nm treatment" the authors explicitly mention "L. donovani parasites counted with an automated cell counter". The determination of cell viability, as described in the title, is only possible if the parasites were previously labeled with a vital dye, and the automated cell counter can distinguish between live and dead cells. Otherwise, the counting is limited to particles of a specific size, without providing information about their viability.

Furthermore, in the Figure 1 caption, the authors indicate: "Parasitic viability of both treated and untreated samples was assessed through microscopic examination and manual counting using a hemocytometer at both 0 and 5 hours post-treatment with 405nm light". The specific instances where each methodology was applied and the criteria for determining parasite viability, such as evaluating factors like mobility, remain unclear.

It is recommended that the authors offer additional details regarding the methodology employed in this particular aspect.

6.       In section 2.7, I recommend that the authors include a bibliographic reference or verify the reliability of the LIVE/DEAD™ Fixable Aqua Stain system for distinguishing live/dead cells in Leishmania donovani. This is crucial as the commercial kit is validated for eukaryotic cells, mainly from mammals. For some protozoa, like T. cruzi, certain vital dyes have proven unreliable due to variations in parasite membrane structures and compositions compared to mammalian cells.

7.       As the manuscript unfolds, it becomes evident that the authors carry out various experiments using platelets spiked with different quantities of parasites per milliliter. In section 2.5, they use expressions such as "low titer (104 parasites/mL)" and "high titer (107 parasites/mL)". However, it remains unclear whether the chosen values align with the expected titers post-isolation of platelets from the blood of individuals infected with Leishmania donovani. The authors are encouraged to provide insights into the reasoning behind their selection of specific parasite quantities and consider including relevant bibliographic references.

8.       It is suggested that the authors resize the figures in accordance with the journal's recommendations and improve their quality to enhance the visualization of the information.

9.       In Figures 1B and 1C, the authors note the presence of a subpopulation of non-viable parasites at low parasite titers, regardless of whether they underwent treatment with 405 nm light. However, the manuscript does not provide conclusions on this observation. It is possible that the sensitivity of the technique could pose challenges in obtaining dependable determinations for titers nearing 104 parasites/ml.

10.    It is recommended that the authors explicitly clarify the meaning of the abbreviations used in the captions of Figures 1 and 2. For example: C - control group.

11.    It is recommended that the authors include the appropriate bibliographic reference that validates the experimental conditions in the Rag2-/- mice infection model with Leishmania donovani.

12.    Incorporating a review of the authors' own publications and those of other researchers in section 4 of the manuscript would be advantageous. This review should delve into the favorable and/or adverse effects on the viability and functionality of platelets following exposure to 405 nm light.

I trust that the authors take into account my recommendations. This manuscript could significantly contribute to discovering an innovative approach for reducing pathogens in transfusable blood products, thereby enhancing safety and storage duration.

Author Response

Thank you for providing detailed comments to improve the manuscript. Please see attached document which provides point by point responses to your comments. 

Reviewer 2 Report

Comments and Suggestions for Authors

Dear Authors,

Although the revised manuscript gives me a feeling of repeatability, with a different target - L. donovani (referring to the articles already published by Michelle Maclean - WOS, PubMed), I believe that the research carried out is well described, the results are clearly stated, the discussions are comprehensive, and the English is impeccable. As such, your manuscript can be published in its current form.

Author Response

Thank you for your positive comments.